# Digital transformation in schools of two southern regions of Sweden through implementation-informed approach: A mixed-methods study protocol

Italo Masiello[1]*, Dean L. Fixsen[2], Susanna Nordmark[1], Zeynab (Artemis) Mohseni[1], Kristina Holmberg[3], John Rack[4], Mattias Davidsson[1], Tobias Andersson-Gidlund[1], Hanna Augustsson[5]

1 Department of Computer Science and Media Technology, Linnaeus University, Växjö, Sweden, 2 Active Implementation Research Network, Chapel Hill, NC, United States of America, 3 Department of Education and Teachers' Practice, Linnaeus University, Växjö, Sweden, 4 Department of Pedagogy and Learning, Linnaeus University, Växjö, Sweden, 5 Department of Learning, Informatics, Management and Ethics, Karolinska Institutet, Solna, Sweden

* italo.masiello@lnu.se

**Data Availability Statement:** No datasets were generated or analyzed during the development of

## Abstract

### Background

The enhancement of–or even a shift from–traditional teaching and learning processes to corresponding digital practices has been rapidly occurring during the last two decades. The evidence of this ongoing change is still modest or even weak. However, the adaptation of implementation science in educational settings, a research approach which arose in the healthcare field, offers promising results for systematic and sustained improvements in schools. The aim of this study is to understand how the systematic professional development of teachers and schools principals (the intervention) to use digital learning materials and learning analytics dashboards (the innovations) could allow for innovative and lasting impacts in terms of a sustained implementation strategy, improved teaching practices and student outcomes, as well as evidence-based design of digital learning material and learning analytics dashboards.

### Methods

This longitudinal study uses a quasi-experimental cluster design with schools as the unit. The researchers will enroll gradually 145 experimental schools in the study. In the experimental schools the research team will form a School Team, consisting of teachers/learning-technologists, school principals, and researchers, to support teachers' use of the innovations, with student achievement as the dependent variable. For the experimental schools, the intervention is based on the four longitudinal stages comprising the Active Implementation Framework. With an anticipated student sample of about 13,000 students in grades 1–9, student outcomes data are going to be analyzed using hierarchical linear models.

this protocol. All relevant data from this study will be made available upon study completion at https://doi.org/10.5878/929p-cf12.

**Funding:** This work is supported by the Swedish Research Council for Health, Working Life and Welfare (FORTE), grant number 2020-01221 and Växjö Kommun, grant number 2020/3209-5.1.1. The funders did not and will not have a role in study design, data collection and analysis, decision to publish, or preparation of the manuscript.

**Competing interests:** The authors have declared that no competing interests exist.

## Discussion

The project seeks to address a pronounced need for favorable conditions for children's learning supported by a specific implementation framework targeting teachers, and to contribute with knowledge about the promotion of improved teaching practices and student outcomes. The project will build capacity using implementation of educational technology in Swedish educational settings.

## Background

The enhancement of–or even a shift from–traditional teaching and learning processes to corresponding digital practices has been occurring during the last 30 years. In Sweden, this process is now called 'digital transformation' of schools [1]. Because of this short evolution time, the evidence of digital technology as an alternative or complementary method for supporting teaching and learning processes is still modest, or even weak [2, 3]; many factors have yet to be considered. It is argued that one of the major causes of this lies in shortcomings in research methods used in this line of research [3], which have been unable to evaluate the promised outcomes through empirical testing. The COVID-19 pandemic disrupted education, for which the aftermath is still being measured, and in order to prevent future disruptions this warrants the need to understand how to prepare teachers to better integrate digital technology into the classroom.

A recent review investigated what barriers and facilitators influence teachers's use of digital technologies in their teaching [4]. The authors identified a reciprocal relationship between school culture and teachers' knowledge, attitudes, and skills. Factors such as the availability of learning opportunities, collaborative learning, a supporting infrastructure, technical support, and a culture where teachers were valued for their work were identified to as positive influences on the implementation of digital technologies.

The Swedish government has laid down a national strategy for digital transformation in schools, with the intention to strengthen both students' and teachers' digital competences and skills [1]. The strategy has brought changes to the national curriculum for primary school level, starting July 1st, 2018. However, the strategy lacks a plan on how to approach these changes, although there are suggested areas on which to focus to move forward [5]. This project links together the digital transformation strategy with the enhancement of digital competences for teachers and students and student outcomes.

Seven core skills make up digital competence: technical, information management, communication, collaboration, creativity, critical thinking and problem solving [6]. Having these skills means, on the one hand, having the technical skills necessary to use digital technology and services and, on the other hand, having the knowledge necessary to find, analyze and critically evaluate information in different media, that is, media and information literacy. Therefore, this project emerges from an educational need, focused on mathematics and literacy. All students will need a basic understanding of numeracy and problem solving as well as literacy since future careers contributing to a sustainable society will require an increasing level of their proficiency in these areas. However, during the last decade, the Swedish school system has been facing challenges in ensuring Quality Education for all students and Reduced Inequalities between students; those are two of the Sustainable Development Goals of the 2030 Agenda that this project covers. PISA has shown that amongst these challenges there has been a declining level of knowledge between 2000–2012 in key skills such as mathematics and

literacy in Sweden, climbing again during the last five years [7]. This was also the case for low performing students, including those with a foreign background (ibid).

Most of today's digital learning materials (DLMs) are curriculum-based and are formed of packages with learning activities consisting of texts, pictures, exercises, videos, and audio. One of the goals of DLMs is to promote learning for all children, especially for newly arrived students whose mother tongue is not Swedish, thus contributing to a more equal and inclusive school. Research evidence indicates that novice readers, who utilize DLM for educational purposes from early age and demonstrate quick and precise word reading skills, do not experience any drawbacks in understanding while reading digitally presented material [8]. This is promising when considering the possibility of new disruptions of education and the fact that digital technology's footprint in education is growing. Digital data from DLMs can be analyzed via Learning Analytics (LA), defined as "the measurement, collection, analysis and reporting of data about learners and their contexts, for purposes of understanding and optimizing learning and the environments in which it occurs" (p. 1) [9], and visualized through Learning Analytics Dashboards (LADs) [10]. The teacher remains central to the process of connecting analyses visualized through the LADs with suitable instructional measures or pedagogical actions targeting individual or groups of students. LA can complement teachers' work by allowing them to inform teaching and learning [11], therefore supporting teachers in their pedagogical activities. According to the 2017 Horizon K-12 report [12], the main goals of using LA in primary and secondary education are to predict student outcomes, implement interventions or adapt the curriculum, and even suggest new approaches to improve student success. This is corroborated by recent reviews for which LADs seem to provide efficient predictive models [13] for retention, study success, test score, drop-out and students' well-being [14, 15]. The COVID-19 pandemic functioned as a catalyst for an exponential growth in the use of digital technology in classrooms, and the creation of data from this; however it is not clear that teachers are able to capitalize on this data and there is, therefore, a need to develop LADs for improving pedagogical decision-making.

In this study protocol, the authors describe an implementation program called *Educational Technology in Schools*. The program is an implementation-informed approach to digital transformation in primary school classrooms. The program is the first of its kind to put an implementation framework into practice in an educational context in Sweden to determine systematically how a program can affect specific teaching practices and student outcomes, contributing both to practice and research evidence. We were able to find only one study [16] that followed an implementation approach to examine the impact of a technology-enhanced professional development model on developing and implementing a math curriculum in U.S. public preschools. The model used professional development sessions and in-class coaching. Children who received the model-supported intervention showed significantly greater gains in mathematical knowledge compared to the control group, with higher implementation fidelity associated with consistently higher scores and greater improvements in children's mathematics achievement. In our program, the implementation process is co-planned and co-designed in a school-municipality-university-company partnership.

## Aim and research questions

The aim of this study is to understand how the systematic preparation of teachers and schools (the intervention) to use DLM and LAD (the innovations) could allow for sustained and innovative use of the innovations, improved teaching practices and student outcomes, as well as evidence-based design of DLM and LAD.

The quality of the outcomes and their sustainability beyond the duration of this project are heavily dependent on implementation fidelity, that is, the extent to which an intervention is implemented as intended. The project has the following research questions:

## Primary research questions

1. What is the impact of the intervention on the use of DLM and LAD (RQ1)

2. What is the impact of the intervention on teaching practices and student outcomes? (RQ2)

3. How does the level of implementation fidelity impact the intervention outcomes? (RQ3)

## Secondary research question

4. What can we contribute to design principles of future DLM and LAD? (RQ4)

## Theoretical framework

Implementation science is a research field that lately has been laying the groundwork for exploring teaching and learning processes in schools [17]. Fixsen et al. [18], has defined implementation science as: "A specified set of activities designed to put into practice an activity or program of known dimensions." The methods of implementation science ensure that education changes are executed to assure that the implementation process accounts for local variables and other contextual factors in schools, in order to scale up and succeed in any setting [19]. The process of implementation unfolds over a series of stages with different implementation activities at each stage. According to the Active Implementation Framework, four stages are common: *Exploration* (assessing needs, creating readiness, etc), *Installation* (selecting and training practitioners/participants, etc), *Initial Implementation* (introducing changes, improvement cycles, etc), and *Full Implementation* (new practices are implemented to a high extent) [20]. Moreover, there are also common components of successfully implemented programs, i.e., *implementation drivers* (competency, organization and leadership) (ibid). The stages and drivers are derived from long lasting successful programs and review of the implementation research literature [18]. The process can span over a period of 2–4 years. The Active Implementation Framework will be used in this project to support the implementation and evaluation of the program.

DLMs and LADs in primary school is a recent phenomenon with which few teachers, students or school principals are familiar. On the one hand, much of the DLMs available in Sweden today are still under development, and the design of a DLM is crucial for its sustained use. According to research [21], there are two major problems with the existing DLMs. First, they are often designed without taking into account how students learn. Usually, developers incorporate too many features that become a distraction, thereby restraining students from recognizing what is relevant for them to achieve the learning goals. Secondly, many publishers of the digital materials simply produce a digital copy of their existing printed counterparts, failing to take advantage of the potential of technology to promote learning (ibid). On the other hand, LADs are increasingly being used in education at all levels, but particularly so in higher education [22]. These tools allow teachers to make informed decisions using visualization methods, facilitate discussions about educational data between teachers and students, and provide real-time information about students' abilities, performance, progress, and results during lessons or lectures [23]. With relevant data literacy skills, teachers can analyze data and identify key features to better understand student needs. Data literacy is the ability to use data for decision

making, including the ability to identify, collect, analyze, and act on the knowledge generated from data [24]. For example, in our own research, we are developing a LAD that allows teachers to interact with multiple visualizations of student data using machine learning algorithms to explore students' learning and activities [25].

## Materials and methods

### Study design

The project uses a cluster quasi-experimental design with schools as the unit. One-hundred forty-five schools are designated as intervention/experimental and 87 as control schools. The project starts with 5 pilot schools to test the entire study protocol, and this should last circa 2–4 years to allow the fours stages of the Active Implementation Framework to run fully. This project is predicted to run for about 10 years, and schools will be enrolled gradually. School principals enroll their school as experimental schools and receive the intervention, which is based on the four longitudinal stages comprising the Active Implementation Framework and its drivers. While Stage 1 is ongoing, the research team can start analysing the data generated by the DLMs before the intervention, allowing to also run a before-and-after study of the experimental schools. In the experimental schools the research team will form a School Team (*the implementation independent variable*) to support teachers' use of DLM and LAD with fidelity (*the implementation dependent variable*) (the School Team is defined in section Stage 1. Exploration, Implementation drivers). DLMs and LADs are the *innovation independent variable* and student achievement is the *innovation dependent variable*. In the control schools, teachers use the DLM but will not have the support from the School Team for using the DLM as intended. Additionally, the control schools will not have access to a LAD.

We will use a mixed-methods (quantitative and qualitative data), and a multi-site and multi-informant (teachers, students, school principals, learning-technologists, sysadmins, and companies' personnel) approach. This study has been approved by the Swedish Ethical Advisory Board (Dnr 2021-06400-01 and supplementary records Dnr 2022-06055-02).

### Recruitment and participants

**Schools.** For the study, we will gradually recruit 145 municipal schools to the intervention part of the study over a period of 10 years. According to the latest statistics (Swedish National Agency for Education, https://www.skolverket.se/) for the school year 2021/2022, there are 232 feasible municipal schools in the two southern regions of Sweden that could participate in the study. A comparison group of schools matched for socio-economic status, location (regional, rural, remote) number of student enrolments and use of the same DLM, will act as controls at the school/unit level. All participating experimental schools will complete a formal contract consent process. Details of requirements for schools to participate in the study is outlined by the research team to the school principal and the school education manager of the municipality in the contract.

The majority of municipal schools in Sweden were already equipped with both IT infrastructure and ICT support staff. However, after the COVID-19, all municipal schools obtain resources from the government to update and renew their IT infrastructure and strengthen the technical support to teachers. Besides, during the pandemic, all Swedish schools were able to use and test all available DLMs in Sweden for free, contributing to new procurements.

**Teachers.** The schools that choose to participate agree to a continuing professional development (CPD) program of digital competences for all teachers, without distinction, except for teachers who specifically decide not to receive any CPD. Teachers have diverse teaching experience and levels of professional independence. Therefore, it is crucial to run pilot studies first

to cater for the appropriate level of CPD activities for teachers. In this case, pilot studies can run for only a smaller group of teachers, e.g., all teachers of students in grades 4–6.

The teacher population in the two southern regions of Sweden is 3,773, of which 75% are women. The student ratio per teacher is circa 11 on average. Moreover, over 88% of the teachers have a permanent employment, either full-time or part-time, and 86% are certified teachers. A simple average across all 232 schools returns circa 2,300 available teachers in 145 municipal schools who can participate in the study.

In the program, the teachers are research partners and guide the direction of the research towards the needs of the teachers and students in the classroom. They are also co-designers of the LADs they are going to use.

**Students.**  Just as is the case with the teachers, pilot studies with a limited student population will be run first to avoid bias and confounding dictated by the different ages. Once the factors influencing the interventions are understood, is it possible to scale up the intervention to all school grades.

The study population is 7–15 years old students (grades 1–9). According to the latest statistics, there are 42,625 students of grades 1–9 in the two southern regions of Sweden, and circa 49% are female. There are 23% of students with a foreign background and 56% with guardians who have post-secondary education in one region, while the other region has respectively 31% and 56%. A simple average across all 232 schools returns circa 26,600 available students in 145 municipal schools who can participate in the study.

**EdTech companies.**  The researchers also set up also a formal agreement with EdTech providers, or companies, that supply their respective DLM to the participating schools. It is important to notice that each provider and service is chosen and purchased by the local municipality, and not the research group. Said formal agreement describes the goals of the program, but more importantly the terms of collaboration between the companies and the university. It also stipulates the sharing of resources. One important term of collaboration is that the participating companies allow a developer and a customer service person to assist the project for a small percentage of his/her full time.

There are technical issues, e.g., sharing data, interpreting the database and format of datasets, as well as supporting the analysis of data, with which the company developers can assist the university. There are also pedagogical issues, e.g., the integration of the DLM into everyday teaching practices and activities and how to perform certain tasks related to learning activities, with which a company-employed teacher with deep knowledge of the DLM can assist the university and the participating teachers during workshops.

The companies are full research partners and have a say on the running of the research work. This gives valuable and direct insight into further DLM development for the participating companies. Moreover, it provides evidence-based support for methods that have been co-designed with and validated by teachers and researchers.

## Procedure

We use the Active Implementation Framework with its four longitudinal stages: *Exploration*, *Installation*, *Initial Implementation*, and *Full Implementation*, and different combinations of its three implementation drivers (competency, organization and leadership) (Fig 1). The process spans over a period of 2–4 years.

The stages and drivers are the following, while data collection and analysis in each stage are presented below:

A.  Implementation Drivers:

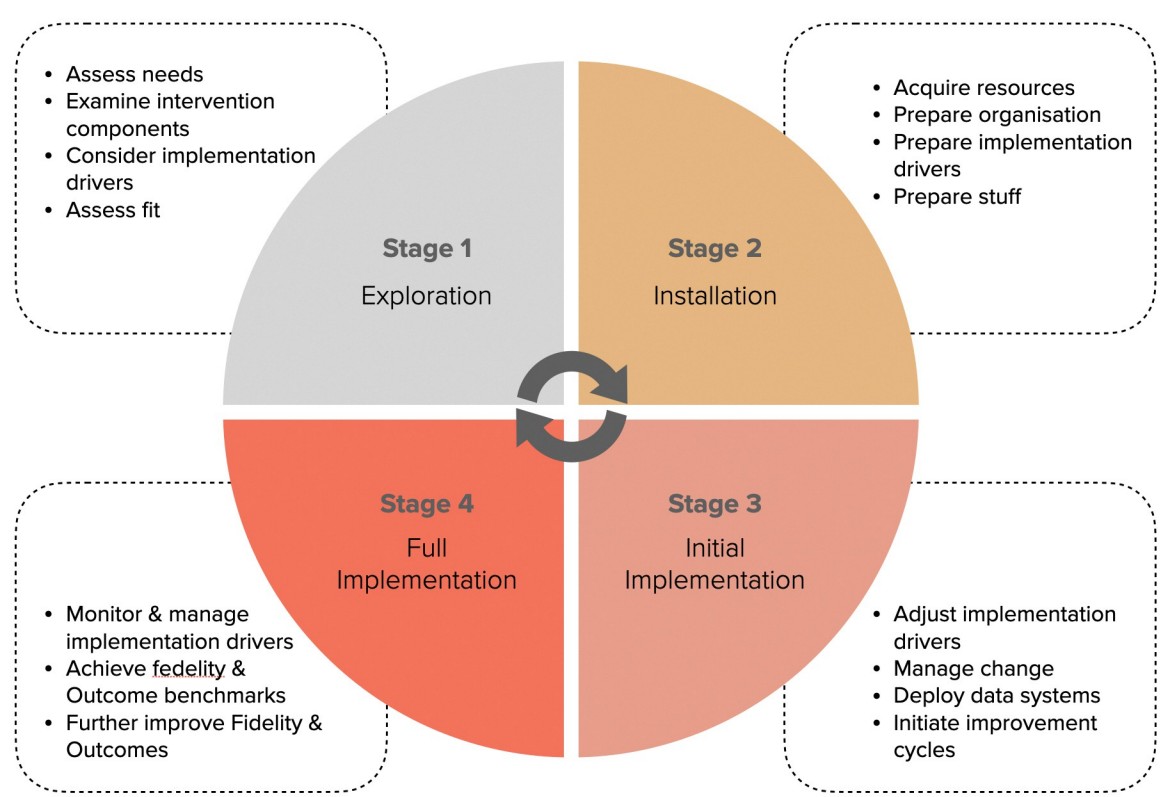

**Fig 1. The four stages of the active implementation framework.**

- Competency drivers: Selection of personnel to run the program, such as School Team members, piloting teachers, education administrators; training program for the School Teams; and researchers having a coaching function.

- Organization drivers: All data systems to be used in the program, such as the DLMs and LADs; the facilitative administrators, such as sysadmin, ICT- instructor; and system intervention, such as new learning activities and curriculum integration activities.

- Leadership drivers: Technical skills necessary for managing daily operation when the program is fully running; and adaptive skills for planning and following up activities with the School Team when the program is fully running.

B. Stage 1. Exploration

- Consider implementation drivers listed in point A.

- Assess needs of teachers, school principals, and companies.

- Examine intervention components: data sources, survey instruments, LADs, STs training, all possible stakeholders; and the Core components: practice profiles, performance assessment (fidelity), impact on teaching practices, impact on student outcomes, organizational development.

- Assess fit by sharing decisions on the program and of the research plan.

C. Stage 2. Installation (the School Teams build the capacity of the program)

- Prepare implementation drivers listed in point A.

- Acquire resources: workshop planning, data sources and storage, material, ethical approval, information consents, lists of participants.

- Prepare organization: build the implementation capacity through meeting with company and school leaders.

- Prepare staff: competence development workshops, planning meetings with municipal system administrators and data-protection ombuds, planning meetings with companies' system administrators.

- Collect research data.

D. Stage 3. Initial Implementation (the School Teams assume a coaching capacity)

- Adjust implementation drivers listed in point A.

- Manage change: finalize technical solutions and dashboards with respective responsible stakeholder, integrate technical solutions, provide evidence-based solutions for further development of DLM.

- Deploy data systems such as LADs and possibly other systems arising from implementation drivers.

- Initiate improvement cycles with teachers and school principals by running short-term studies to test research hypotheses based on data collected.

- Measure fidelity by looking at impact on teacher practices and student outcomes.

E. Stage 4. Full Implementation (the School Teams assume a helping capacity)

- Monitor and manage implementation drivers listed in point A.

- Achieve fidelity and outcomes: integration of new practices and LADs for the majority of the teachers and school principals. Achieve students results through the new practices.

- Further improve fidelity and outcomes by drawing an implementation plan and creating a learning collaborative.

## Data collection and analysis

**Stage 1. Exploration.** The main aim of the Exploration Stage is "[. . .] to consider the extent to which a potential innovation or approach meets the needs of the community, and whether implementation is feasible" [26]; in other words, to determine which practice or program is the best fit by examining needs and assets, and preparing the organization for implementation readiness, through the efforts of the School Team. Community planning is important to achieve sustainability of high-quality implementation [18]. Successful and rigorous planning at this stage will determine the outcomes of the intervention in the next three stages. By using mixed methods [27], the researchers will prepare the ground for implementation. Right at the onset, the researchers will email all school principals about the projects and the need to develop an infrastructure for implementation. This will invite school principals to participate in the intervention program. With the participating schools the research team will draw a memorandum of understanding with clear values and a vision for the project, objectives and intended outcomes.

*Implementation drivers. Competency drivers*: *Formation and training of the School Team(s).* To work with the experimental schools, we put together and train a School Team for each municipality. According to the literature [28], expertise is necessary for the deliberate and efficient application of the AIF, ensuring that innovations consistently yield desired results in a reliable and repeatable manner. This expertise is cultivated within implementation teams established in each organization, and their impact on implementation outcomes has been proven [29]. In our program, the implementation team is called the School Team. Each School Team can consist of two researchers, one or two specially appointed teachers, one of which could be also a learning-technologist and/or school principals, and one representative from the EdTech provider. When the project will scale up, more than one School Team is going to be needed per municipality. The School Team is the implementation independent variable that supports teachers' use of the DLMs and LAD with fidelity (the implementation dependent variables).

*Assess needs. Interviews.* Through focus group interviews with teachers at each school, we want to assess the needs and better understand how teachers and school principals relate to and use digital technology in the classroom, as well as in the school at large. Raw data from the interviews is analyzed with qualitative content analysis [30].

*Assess fit.* All the School Teams gather and meet periodically in a general meeting to discuss the progress of the program, general issues, follow-up the activities, and adapt and plan (*Plan Do Study Act*—PDSA cycles). It is during these meetings that all members have the possibility of assessing and determining program fit and to re-design activities if necessary.

*Examine intervention components.* It is also during the School Teams general meeting that the members review and discuss all the instruments and protocols used in this program and how to distribute them.

*Core components.* The term core components refers to the essential functions, principles, associated elements and intervention activities that are necessary to produce desired outcomes [26]. The core components of the program are the following:

- Impact on teacher digital practices (implementation dependent variable) on the use of DLMs and LAD.

  ○ through School Teams work (Implementation independent variable), workshops, assessment of *practice profiles* and observations.

- Impact on school organizations and EdTech providers (innovation dependent variables).

  ○ through DLM and LAD use (innovation independent variable).

  ○ through school principal's LAD consultations and integration.

  ○ through school principal's decision-making activities related to LAD consultations.

  ○ through EdTech providers' implementation of new DLM features in connection with workshop data and DLM data.

  ○ though dialog and interviews with the goal of understanding the impact for stakeholders.

- Impact on student results (innovation dependent variable).

  ○ through use of DLMs and consequent teacher decision-making by LAD consultation.

- Systemic, sustainable implementation fidelity.

  ○ through sustained implementation strategy of the program reform and improved teaching practices (implementation dependent variables).

*Practice profile*. The Practice Profile, referred to as innovation configurations [31], describes the essential activities that permit the innovation to be teachable, learnable, and doable in typical service settings, allowing consistency across staff. In this program, the Practice Profile for teachers is related to a set of measurable behaviors that demonstrate use of digital competences through the program's activities. Therefore, the Practice Profile is connected to the educators' pedagogical competencies shown when using the DLMs and LAD. Practical assessment of those behaviors is made using, for example, observations, logbooks, and surveys to collect information about whether a teacher uses a DLM for assessment or collaborative learning activities and to what extent.

*Fidelity*. To assess fidelity, we create a digital logbook for teachers, run classroom observation sessions and collect implementation fidelity questionnaires. Outcomes from all three data collections are then correlated to determine how reliable they are and to provide an overall assessment of fidelity. The assumption is that higher fidelity will be associated with better outcomes.

All behaviors measured through the three methods are connected to the educator's digital competences explained in section Stage 2. Installation, under SELFIE for teachers, and making up the Practice Profile.

*Digital logbook*. All participating teachers will fill out a digital logbook to assess when, how and for what purposes they use a DLM and LAD, or if other digital resources are used. The digital logbook uses different scales and short-answer survey questions and teachers will self-mark which behaviors they have adopted at intervals between two and four weeks.

Descriptive statistics are used to calculate the self-scoring scores and content analysis for the survey questions.

*Classroom observations*. The Digital Logbook is a self-assessment instrument that is complemented by objective, researcher-driven, classroom observations. A researcher uses an observation protocol built on the Practice Profile's behaviors, as explained above. The observations will objectively mark the development of teachers' educational practice, both when they use a DLM first and a LAD later. The researcher's objective observations add to the level of evidence of the fidelity to the core components of the program. The observation protocol is designed with a four-point scale behavioral measure, and descriptive statistics are used to calculate the observation scores.

*S-NoMAD instrument*. To assess the implementation processes and understand how the intervention in our program works and how it can be improved, we make use of the Swedish version of the instrument Normalization Process Theory Measure (NoMAD), S-NoMAD [32]. NoMAD has been translated into Swedish and its psychometrics properties validated through confirmatory factor analysis of the four constructs making up S-NoMAD: *Coherence*, *Cognitive Participation*, *Collective Actions* and *Reflexive Monitoring*. *Coherence* relates to the work that people do individually or collectively to implement new practices; *Cognitive Participation* is the work that people do to build relations to sustain a community of practice; *Collective Actions* is the work that people do to enact a set of practices; and *Reflexive Monitoring* is the work that people do to assess and understand the ways that a new set of practices affect them and others (ibid).

S-NoMAD's 23 items, with a five-point Likert scale, provide possibility for adaptation for specific context and study protocol. The collection of the S-NoMAD questionnaire should be carried out at the end of Stage 2. Descriptive statistics are used to calculate the item and scale scores. Internal consistency of the total score and the four theoretical constructs is analyzed by calculating the Cronbach alpha [33] for the pooled dataset.

The result of this stage is a clear implementation plan with tasks and timelines to facilitate the next two stages.

**Stage 2. Installation.** The Installation Stage is "[. . .] when needed organizational and personal competencies are established to ensure the successful implementation of the selected innovation" [26]. In the following sections, we outline how Stage 2 should be realized within this specific program.

*Implementation drivers. Competency drivers.* During Stage 2, the School Teams build the capacity of the program, and the members are involved in all activities, from distributing informed consent forms to providing the competence development to the teachers.

*Organization drivers.* The School Teams are instrumental in distributing and retrieving the signed informed consent forms from the legal guardians of the students and generate lists of all people (except for students) involved in the program. The researchers set up discussion meetings with sysadmins of the municipalities, those in charge of the students' data, sysadmins in the EdTech companies, and data protection officers in the municipalities, to reciprocally learn about the data that is available, what parts of the data collection that are most challenging, all the ethical aspects related to students' data, and, finally, what do each of them want to obtain from participating in this program and collaboration.

*Acquire resources.* One of the important resources in Stage 2 is to plan and develop workshops for professional development of the participating teachers' digital competence. Each school sets up a schedule fitting their context and conditions. In addition, we acquire and document informed consent orally from all participating teachers and students, and signed informed consent from all student guardians. Only when all consent matters have been cleared can the researchers start collecting and analyzing digital data related to the students, explained below.

*Learning analytics dashboard.* During the last part of Stage 2, the researchers, in strict collaboration with the users, will develop the LADs that will be used by teachers, school principals, and participating companies. Data from the DLMs and from the municipality will be merged and hashed so that the researchers will only work with pseudo-anonymized data. Pseudo-anonymization is the process of handling personal data in a way that makes it impossible to link it to a specific data subject without the aid of additional information [34]. In our case, and for obvious educational reasons, it is only the teachers and the school principals who have the key to determine who the individual students are, while the researchers only work with anonymized data. Data is pre-processed and analyzed by machine learning (ML) algorithms to find patterns and is visualized by visualization techniques.

During the pilot period with five schools, the researchers will co-design and use design-based research [35] combined with a series of workshops with all participating teachers to develop a viable LAD that will then be finalised durign Stage 2, as explained above. The results of the workshops will help us to design paper prototypes, and subsequently clickable prototypes to better understand teacher needs. This process will be iterated multiple times (usually three or four). The LAD for teachers will be tested during Stage 2 and the final version used during Stage 3 of the implementation.

*Prepare organization. SELFIE.* The significance of digital capacity and maturity in educational organizations has grown in recent years [36]. Organizational digital capability refers to the extent to which an organization's culture, policies, and infrastructure enable and support the use of digital technology [37], while digital maturity refers to the systematic use of technology in school management and teaching practices [38]. To understand those in relation to our program, we first collect and analyze the municipal strategies. Then, we measure digital capacity and maturity by collecting a self-reflection instrument developed by the European Commission with a panel of experts from across Europe, called SELFIE: *Self-reflection on Effective Learning by Fostering the use of Innovative Educational technologies* [39].

SELFIE comprises a questionnaire in three parts, for school principals, teachers, and students, on a five-point Likert scale. We will aggregate the data from the three stakeholders per individual school to describe the whole school and perform K-Means Cluster analysis with aggregated data. We will run a regression analysis for each cluster.

*Prepare staff. SELFIE for teachers.* 'SELFIE' is the EU instrument designed for self-reflection of the digital capacity and maturity of educational institutions, in our case primary schools. 'SELFIE for teachers' (https://education.ec.europa.eu/selfie-for-teachers) is instead meant to help teachers understand and develop their own digital competences, specifically related to their use of digital technologies in their work with colleagues, learners, and others. The DigCompEdu Framework (European Digital Competence Framework) was developed by the European Commission's Joint Research Centre on behalf of the Directorate-General for Education, Youth, Sport, and Culture, and is intended for use by educators at all levels, including early childhood education, higher education, vocational training, special needs education, and non-formal learning contexts. The framework proposes 22 elementary competences organized into six areas (Fig 2): Area 1) the broader professional environment; Area 2) the creation and use of digital resources for learning; Area 3) the management and orchestration of digital technologies in teaching and learning; Area 4) the use of digital strategies for assessment; Area 5) learner-centered teaching and learning strategies using digital technologies; and Area 6) specific pedagogical competencies related to facilitating student digital competence. The framework also includes a progression model with six stages to help educators assess and improve their digital competence: Newcomer (A1), Explorer (A2), Integrator (B1), Expert (B2), Leader (C1), and Pioneer (C2).

In this program, we will focus on the development of the educators' pedagogic competencies, Areas 2–5, since we believe that those areas are most pertinent to the teaching and learning activities of everyday classroom practice. The Practice Profile's behaviors build on the competences of areas 2–5. Our effort will focus on bringing all the participating teachers to at least the B2 level, i.e., expert.

Data will be analyzed by descriptive statistics to present teachers' digital competence, ANOVA of teachers' profiles to investigate possible differences in digital competence across profiles and time, and simple and multiple regression analyses to evaluate the effect of teachers' digital competence across schools.

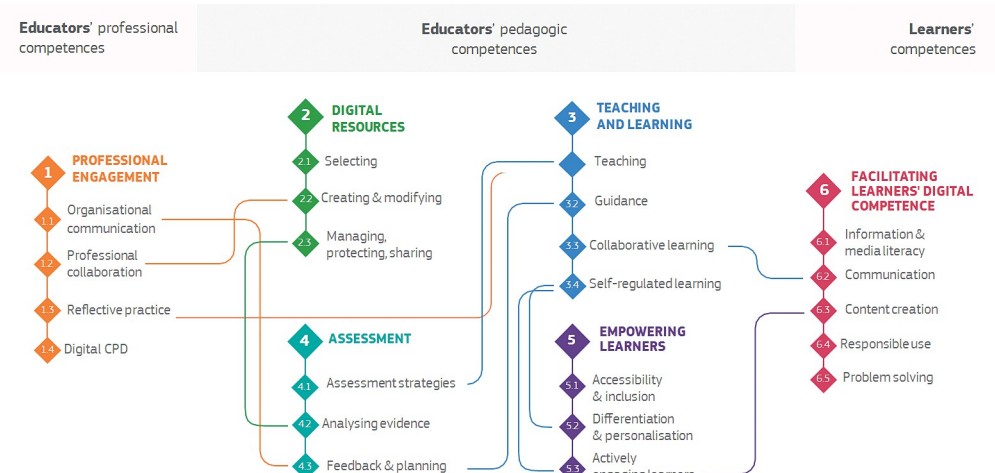

**Fig 2. The 22 digital competences of the DigCompEdu framework.** The use of the figure is authorized by the EU (https://ec.europa.eu/jrc/en/digcompedu).

*Workshops*. All the different steps in Stage 1 and 2 lead to the creation of CPD workshops for the participating teachers. The workshops are both physical and online, and follow different program's formats dictated by the conditions of each school and run for the whole duration of Stage 2. PDSA cycles allow to adapt the content of the CPD workshops to the specific needs of the teachers for greater impact. This means teaching about a specific content, reflecting on a subject or hands-on developing of specific matters together with the companies, so that teachers can get help to fit a specific subject of the DLM into classroom activities. This also means that the pace, collection of research data, and achievements of the research and development work may differ for each participating school.

The workshops are evaluated by continuous dialog and note taking during the workshops, and by formal interviews, analyzed by content analysis. The result of this stage is a developed implementation capacity in each participating school belonging to the experimental group.

**Stage 3: Initial implementation.** "During the Initial Implementation Stage, the new practice is first put into place and made available to consumers. The key focus of this stage is on continuous improvement." [26]. The Initial Implementation Stage is also the most vulnerable stage, i.e., prone to fail, since it is here–when users start using and leaving their comfort zone–that the risk of going back to what is known, and where one feels confident, is at its highest [ibid]. Therefore, the School Team is imperative in 'keeping up the work' and inspiring their colleagues.

*Implementation drivers*. During this stage, the implementation drivers are designed to develop competence and confidence of teachers, create a positive working environment, and provide leadership which can support challenges during the system change.

*Competency drivers*. When the teachers first use the newly learned skills, support for change is essential. During Stage 3, the School Teams help the teachers adjust to the new teaching functions/activities so that they can begin supporting the use of their DLM and LAD. The School Teams hold a coaching function.

*Organization drivers*. The School Teams also help school principals and administrative staff make possible changes in operations of the schools to support the teachers' use of the DLM and LAD as intended–with fidelity. This means enhanced integration of the DLMs and LAD within existing systems improved time planning regarding competence development for new teachers or other staff.

*Leadership drivers*. The researchers start to work with the school principals. Meetings are scheduled for discussing data analysis and the significance of the LADs. Data literacy is equally important for school principals as it is for the teachers. Data-informed principals can identify key features of a LAD and better understand the needs for improvement work based on evidence. The meetings are also necessary to plan for the sustainability of the project after the research program is concluded, that is, creating an initial draft of a Sustainability plan.

*Initiate improvement cycles*. In Stage 3, the researchers conduct before-after experimental studies in the form of improvement cycles. At this point, the participating teachers, scaffolded by the School Teams, should be able to use their DLM and LAD with more determination. Large amounts of digital data are generated, and the LAD should be able to provide information about individual students and/or groups of students who might need attention from the teacher. The teachers are instructed to share that information with the researchers. Vice versa is also true, the researchers monitor the LAD and inform the teachers that attention about individual students or groups of students is needed. Informed by retrieved data, the researchers and the teachers decide on which pedagogical direction to take. This is documented by the researcher by observations in the classroom and observations in the LAD. After a specified time, the researcher will analyze the data to determine if a change has occurred. The researchers can run several such studies to collect evidence of pedagogical-informed actions.

*Measure fidelity*. The School Teams build on possible implementation strengths by observations in the classroom and interviews with the teachers and students as testimonials from those who have experienced success with the intervention. The School Teams conduct planning for PDSA cycles by using the aforementioned methods and the S-NoMAD instrument. This constitutes research evidence that will form the basis of the Sustainability Plan (see the following section). Finally, we interview the school principals and EdTech providers about the DLM and LAD and their functionality connected to usage and learning to understand also the level of fidelity. The data collected is analyzed as explained earlier.

*Manage change*. It is during Stage 3 that we should recognize the development of teachers' digital competences and subsequent student outcomes. The scaffolding provided by the School Teams in terms of a long professional development work of teachers and systematic work of the intervention should be leading to evidence-based improvements. Therefore, one very important evidence is the comparative measure of change between the experimental and the control schools.

School level aggregated data about the use of the DLM and about the students is collected from all schools, i.e., experimental and control, within the two southern regions, as well as from the EdTech companies and the municipalities. This data is rendered anonymous and averaged on school level. The researchers will answer RQ1 and partially RQ3 by running comparative statistical analysis of aggregated data of two sorts: a) compare the use of the DLM between schools, expecting that the experimental schools will see a higher degree of use; and b) compare students from the same grades between schools, expecting that students in the experimental schools will see a higher degree of use.

SELFIE and SELFIE for teachers will also provide a comparative measure, this time within schools. In this case, both instruments are self-reflective on the practice of educational technology in schools but also on the development of digital competences. The researchers will answer partially RQ1, RQ2 and RQ3 by running comparative statistical analysis of two sorts: a) compare change between the teachers, students, and school principals receiving the intervention, and teachers and students on an aggregated level not receiving the intervention within the same school because they opted to not participate; b) compare changes (before-after) in digital competence development level of each teacher receiving the intervention.

Another change is that supported by the use of the LAD in the experimental schools and on individual students. The sustained implementation of the LAD and its pedagogical interpretation can be measured by experimental studies by comparing before and after a pedagogical action, as explained earlier (answering RQ2). Students are organized within classrooms; therefore, student outcomes data are going to be analyzed using hierarchical linear model. The hierarchical model accounts for both student- and classroom-level variability in outcomes with a two-level model (nested) [40].

The result of this stage is measurement of innovation fidelity, improvement cycles, and data on student outcomes. These results are important to further develop the DLM together with the EdTech companies, also providing an answer to RQ4, improvement work in schools together with principals, better integration of the DLM in the school, possible integration of the LAD in the DLM, and further develop the LAD.

*Deploy data systems*. As stated above, the results in this stage are important for the further development of the DLMs (answering RQ4) and LAD. Moreover, each municipality should decide how to best integrate these data systems in the schools. This information is also part of the Sustainability Plan. If other technical systems are going to be necessary for better integration of the DLMs and LAD, those systems will be discussed at this stage.

**Stage 4: Full implementation.** Once the new practice is assimilated by teachers, school principals, and students then full implementation can be reached. Full implementation means

the intervention is fully operational, teachers can use the DLM and LAD with fidelity, school principals and administrators support and facilitate the new practices, and students have adapted to the presence of the DLM as a complementary learning material.

*Implementation drivers*. "Continuous quality improvement of the Implementation Drivers is a hallmark of Full Implementation" [26]. At this stage the School Teams monitor and manage the implementation drivers.

*Competency drivers*. The support of the School Teams is still present as a helping capacity, since it is imperative that new staff get trained and 'buy in' so that the program runs according to the planned criteria. During Stage 3, the intervention can scale up to include new schools, why it is necessary to involve new members in the School Teams, or even to set up new School Teams altogether.

*Organization drivers*. The use of the DLMs and LADs becomes standard practice, and they are well integrated into the technical solutions of the school, with a clear administration and management plan.

*Leadership drivers*. The school principal has been trained by the School Team and has the necessary technical skills to reap the benefits of the LADs for improvement work and to manage and follow up the sustained operation of the intervention on a school level.

*Achieve fidelity*. At this stage the School Team will work on innovation and sustainability, creating a learning collaborative, i.e., a professional learning community providing ongoing training, such as introducing new content over time and extending intentions to other teachers and subjects. The goal is to have desirable changes at an implementation school included as the 'standard program'.

The school principal plays a crucial role in creating the conditions for sustaining fidelity of new practices and student results.

*Further improve fidelity and outcomes*. After establishing a fully implemented evidence-based program, the program needs to be sustained beyond the period of the project. Thus, during this stage a Sustainability Plan will be drafted in collaboration with the respective School Team to ensure that the school can tackle any changes if staff, including champions and leaders, leave, or if curriculum changes occur. Discussion will be initiated with other municipality schools that still have not participated in the intervention, but already have access to the DLM, those who participated as control schools, and with entirely new units, to provide scaling up to a larger population. The result of this is a fully implemented evidence-based program and Sustainability Plan.

## Sample size calculation

The study uses a cluster quasi-experimental design with schools as the unit. There are 232 municipal schools in the two southern regions of Sweden that make up the school population. With precision level at ±5%, a confidence level at 95% and standard deviation at .5, the sample size to reach statistical significance is 145 schools, designated as intervention/experimental, and 87 as control schools. That also provides an average student population of about 26,600.

## Data management

A data management plan that ensures and maximizes the openness, findability, exploitation, and good management of all data generated and handled in the project will be necessary for this program. The researchers collect aggregated data about teachers and classrooms without infringing any professional, data management, or ethical regulation. The researchers also collect data about implementation, which follow the same principles. However, we collect digital data about individual students, which falls under different ethical regulation. In this case, we

will apply the FAIR principles (https://www.go-fair.org/fair-principles/) and make available unidentified user data through the Swedish repositories DORIS (https://doris.snd.gu.se/), so that other researchers can validate, reproduce, or reuse the research outputs. The project leader coordinates and monitors the process.

## Discussion

This study protocol introduces a quasi-experimental cluster study addressing the conditions for children's learning in school and focuses on optimal strategies that affect implementation. To address the weakness of a non-randomized design, we also run a before-after study of the experiemental schools. We are aware of the fact that the outcomes may be skewed as a result of postive factors associated with schools that decide to receive the intervention. On the other hand, demonstrating impact could be harder since all schools in Sweden are looking at ways to improve teachers' digital competence; and our program provides systematic professional development that could make the use of DLMs and LADs in the classroom more effective. School activities are complex: they are fast changing because of the affordances provided by educational technology, they face many problems with keeping competent teachers, and they follow ambiguous evidence.

The project makes two main contributions: one to society, and one to the research on CPD of teachers to build implementation capacity. The project contribution with societal relevance consists in targeting a large professional group that represents a basic social function, teachers. Implementation science has proved theoretically and practically to support teaching practices and student outcomes, yet new to Swedish schools. The project is expected to result in both short-term and long-term positive development of teaching practices and student outcomes. These are of great societal value, as they lead to professionalism and employability. The project is also expected to influence the evidence-based design of DLMs used in the participating schools.

The intervention program does not require major organizational changes, as all Swedish teachers have several hours per week for personal development in Sweden. The program is therefore expected to be favorable to implement, both during the project and beyond. As the project includes ongoing implementation initiatives of integration of digital competence, the outcomes should provide favorable conditions for continued implementation. Therefore, the contribution to research on systematic CPD of teachers is expected to enhance knowledge about the promotion of a sustained implementation strategy to improve teaching practices and, subsequently, student outcomes. We expect that once we have analyzed all data available from the different sources and the co-created LADs that teachers can use, we can create data literacy workshops for the teachers. The benefit of data-informed practices through the LADs requires to be demonstrated and explained to teachers, for them to realize the overall advantages. The role of the School Teams will be to aid teachers in understanding how to translate and realize this knowledge into real pedagogical interventions, in order to make effective pedagogic use of the LADs benefitting their students.

In summary, the program aims to address a pronounced need for favorable conditions for children's learning by using educational technology with the help of a specific implementation framework targeting teachers. The contributions of our study is expected to increase the attractiveness of the teaching profession, as well as the satisfaction of parents and children.

## Author Contributions

**Conceptualization:** Italo Masiello, Dean L. Fixsen, Susanna Nordmark, Zeynab (Artemis) Mohseni, Kristina Holmberg, John Rack, Mattias Davidsson, Tobias Andersson-Gidlund, Hanna Augustsson.

**Funding acquisition:** Italo Masiello.

**Methodology:** Italo Masiello, Dean L. Fixsen, Susanna Nordmark, Zeynab (Artemis) Mohseni, John Rack, Mattias Davidsson, Tobias Andersson-Gidlund, Hanna Augustsson.

**Project administration:** Italo Masiello, Susanna Nordmark, Kristina Holmberg.

**Resources:** Italo Masiello.

**Supervision:** Italo Masiello, Dean L. Fixsen, Kristina Holmberg, John Rack, Mattias Davidsson.

**Writing – original draft:** Italo Masiello.

**Writing – review & editing:** Italo Masiello, Dean L. Fixsen, Susanna Nordmark, Zeynab (Artemis) Mohseni, Kristina Holmberg, John Rack, Mattias Davidsson, Tobias Andersson-Gidlund, Hanna Augustsson.

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
