## [Decision Letter · Decision Letter 0]

10 Nov 2023

PONE-D-23-04795Digital Transformation in Schools of two South Regions of Sweden through Implementation-Informed Approach: A Mixed-Methods Study ProtocolPLOS ONE

Dear Dr. Masiello,

Thank you for submitting your manuscript to PLOS ONE. After careful consideration, we feel that it has merit but does not fully meet PLOS ONE’s publication criteria as it currently stands. Therefore, we invite you to submit a revised version of the manuscript that addresses the points raised during the review process.

You will note that both reviewers have suggested substantial changes to improve the clarity of the protocol. Please do attend to these.

We look forward to receiving your revised manuscript.

Kind regards,

Subhashni Taylor, PhD

Academic Editor

PLOS ONE

Journal Requirements:

4. We note that the original protocol that you have uploaded as a Supporting Information file contains an institutional logo. As this logo is likely copyrighted, we ask that you please remove it from this file and upload an updated version upon resubmission.

Reviewers' comments:

Reviewer's Responses to Questions

**Comments to the Author**

1. Does the manuscript provide a valid rationale for the proposed study, with clearly identified and justified research questions?

Reviewer #1: Partly

Reviewer #2: Yes

2. Is the protocol technically sound and planned in a manner that will lead to a meaningful outcome and allow testing the stated hypotheses?

Reviewer #1: Partly

Reviewer #2: Yes

3. Is the methodology feasible and described in sufficient detail to allow the work to be replicable?

Reviewer #1: Yes

Reviewer #2: Yes

4. Have the authors described where all data underlying the findings will be made available when the study is complete?

Reviewer #1: Yes

Reviewer #2: Yes

5. Is the manuscript presented in an intelligible fashion and written in standard English?

Reviewer #1: Yes

Reviewer #2: Yes

6. Review Comments to the Author

You may also provide optional suggestions and comments to authors that they might find helpful in planning their study.

Reviewer #1: Thank you for the opportunity to peer review this protocol. Some of my comments are suggested alternative ways of organising some of the work. These are only offered as suggestions that authors might consider, not as requirements for change.

Understandability

- I have read the protocol several times and it still feels quite unclear to me. Authors should attempt to make the intended activity easier to grasp for an external reader, perhaps through a visual flow chart of the activity that is planned and indicating which stakeholders are involved when and how. Because this is a large project over many years, it might be helpful to break down the work into smaller work packages, with specified input/output. Authors should consider creating a logic model, that ties the innovations, intervention and related activities more clearly to the “core components” (intermediary effects?) and desired outcomes.

- Practice profile: I don’t understand what this is.

What is already known/prior research

This protocol lacks description of what we know from earlier research about interventions to improve uptake of digital technology. The questions I had after reading this several times were:

- Why should teachers be using DLM (other than the government having decided that)? There may be many good reasons, but these are not presented.

- Why should teachers be using LADs (other than this research proposal having been funded to implement these)?

- What do we know about why this behaviour change in schools not been happening (is it lack of motivation, capability, opportunity?) What do we know about what teachers think (about the usefulness, barriers, facilitators, of using digital materials in teaching)?

- Where did the idea of School Teams intervention come from? Has it been tried elsewhere in the context of research (if so, has it been proven to be effective?)

- What do we know about other interventions to improve digital technology uptake in the classroom that could be alternatives to School Teams? (Is there a systematic review? Have authors reviewed the primary literature?)

Quasi-experiment vs. randomisation

- In this study, schools would be self-selecting to receive the intervention, rather than randomized. This can have an effect on the results in the favour of the intervention. For instance, schools that are already motivated will be more likely to participate, and they would then be compared to schools that were not so motivated to use digital technology. Authors should acknowledge the weaknesses of a non-randomised design and describe what measures they will take to compensate. (e.g. consider before-after or time-series analysis of intervention and control group). Additionally, final results may be less likely to be applicable to schools that are not already motivated, and therefore do not sign up to participate – this should be discussed.

LAD co-design/development/evaluation

- To an outsider, it seems to me that DLM is a very broad term covering any learning material that is digital, while LAD is a specific product or solution that exists in a beta form (i.e. described “SAVis: a Learning Analytics Dashboard with Interactive Visualization and Machine Learning”). Is that correct? It needs clarifying.

- It is not clear to me why development of LAD would be done within a quasi-experiment design, and the LAD described in the article “SAVis…” appears to not be developed to the point of being suitable for use by people who are not digitally savvy (it looks like a tool for computer experts, not for teachers who are not using much digital technology). Before embarking on a multi-school experiment, my preference would have been to establish a preliminary “development” work package for LAD that included:

o Needs assessment regarding LAD

o Design development: Co-design with groups of key stakeholders (e.g. teachers, principles, school ICT administrators, maybe students and parents too?) involving several cycles of rapid iteration, user testing, and improvement of the LAD, to ensure that the solution is useable, useful, accessible, desirable, and suited/fit for use in schools’ teaching and learning environments.

o Piloting of LAD use by teachers and principles, with data collection (e.g. observation, interviews, logs)

o Assess the effect of LAD on teachers/school principles use, satisfaction, and on student outcomes if possible,

by conducting a randomised trial , to understand if this innovation has the intended effect, before implementing it broadly. (You should be able to show that this LAD product has the desired effect on a set of outcomes, before setting about to implementing it on a broad scale).

School team intervention – development and evaluation before implementation?

- It might be prudent to carry out a preliminary study as I have suggested above (co-development of the School Team intervention by a group of pilot schools, followed by assessment of effect using randomized trial methodology), before seeking to implement this intervention on a broad scale in a non-randomised experiment. Perhaps authors have intended this, but I have not understood it correctly.

Reviewer #2: This paper describes a protocol that will be implemented in two areas of Sweden to document their school's digital transformation. The planned process has been thoroughly described and the research questions are clear and should provide comprehensive findings. The potential number of participants is quite large and, as such, will require detailed procedures such as outlined here.

While the protocol is well-considered, there are a number of general comments that need to be made. Overall the literature cited is dated. For example, the research that refers to 'the existing DLMs' was published in 2012. This is quite a long timeline for something as dynamic as technology implementation.

The protocol makes little reference to the technology infrastructure roll out. I know little about the Swedish school system and it may be already very well resourced in this area and this is already in place? 'Stage 2: Installation' makes the only reference to 'Acquire resources' but there seems little mention elsewhere of infrastructure and specialist technical staff. These would be things that are likely to have a huge impact on the progress of the digital transformation of a school. There is a statement made that 'the intervention program does not require major organizational changes' so perhaps infrastructure and appropriate staffing is already in place.

I wonder if any provision has been made for cross-school comparisons when each local municipality is responsible for the purchase of resources? Will all schools be equally well-resourced in this case and how might any variations affect the research results?

I also note that the EdTech 'companies are full research partners and have a say on the running of the researach work'. This will have to be managed carefully to ensure there is no conflicts of interest.

This will be a significant project and the results will potentially have wide-ranging impact. I look forward to reading about it further in the future.

7. PLOS authors have the option to publish the peer review history of their article (what does this mean?). If published, this will include your full peer review and any attached files.

Reviewer #1: No

Reviewer #2: **Yes: **Leanne Cameron

---

## [Author Response · Author response to Decision Letter 0]

22 Nov 2023

Dear Editor,

Thank you for the opportunity to revise the manuscript. We have attended to all comments raised by you and the reviewers. In this document we have included the comments raised by the editor and reviewers in normal text and responded to those in italics and blue directly under each point.

We have changed the title, headings, figure legends and citations according to the style requirements of the Journal that the Editor has linked to.

We have corrected this.

We have now created the repository and received a DOI number that we share here. We have not uploaded any documents yet and have not completed the repository with all the different instruments we are collecting. We have been in touch with our university responsible person for the repository to plan how to do this correctly as we have sensitive data that we cannot share, and we want to make sure that we do it without infringing on the GDPR. The DOI link is the following: https://doi.org/10.5878/929p-cf12 but since the DOI is not been assessed by the regulatory body yet, use this link instead to see the repository https://snd.gu.se/sv/catalogue/dataset/preview/0b77896b-e5aa-40a2-b117-5896bd6d51f6/1 The page is only available to you with the link. On page 30, we have simply added the name and link to the Swedish repository where the data are going to be available in the future.

4. We note that the original protocol that you have uploaded as a Supporting Information file contains an institutional logo. As this logo is likely copyrighted, we ask that you please remove it from this file and upload an updated version upon resubmission.

We are not sure which document the Editor is referring to. Usually the logos are permitted to be used without infringing on copyrights.

Reviewers' comments:

Reviewer #1: Thank you for the opportunity to peer review this protocol. Some of my comments are suggested alternative ways of organising some of the work. These are only offered as suggestions that authors might consider, not as requirements for change.

Thank you for your suggestions. We have taken into consideration all of the suggestions provided by the reviewer, and addressed most of those but also rejected some other suggestions. The explanations are presented below.

Understandability

- I have read the protocol several times and it still feels quite unclear to me. Authors should attempt to make the intended activity easier to grasp for an external reader, perhaps through a visual flow chart of the activity that is planned and indicating which stakeholders are involved when and how. Because this is a large project over many years, it might be helpful to break down the work into smaller work packages, with specified input/output. Authors should consider creating a logic model, that ties the innovations, intervention and related activities more clearly to the “core components” (intermediary effects?) and desired outcomes.

We have created a logic model, which resulted in a complex picture because of the many parts of the program. The logic model is now under review in another publication, so we have not included it in this submission in order to avoid publication/copyright issues. We are attaching a picture of the logic model here for information.

- Practice profile: I don’t understand what this is.

We have added a couple of sentences with an example in the section Practice Profile to hopefully clarify the concept of Practice Profile. Page 18.

What is already known/prior research

This protocol lacks description of what we know from earlier research about interventions to improve uptake of digital technology. The questions I had after reading this several times were:

- Why should teachers be using DLM (other than the government having decided that)? There may be many good reasons, but these are not presented.

We have added a few sentences on this on page 5 and 6. 

- Why should teachers be using LADs (other than this research proposal having been funded to implement these)?

We have added two sentences on this on page 6. 

- What do we know about why this behaviour change in schools not been happening (is it lack of motivation, capability, opportunity?) What do we know about what teachers think (about the usefulness, barriers, facilitators, of using digital materials in teaching)?

We have added a paragraph on barriers and facilitators on page 4.

- Where did the idea of School Teams intervention come from? Has it been tried elsewhere in the context of research (if so, has it been proven to be effective?)

We have added two sentences on the school team and its impact on page 15.

- What do we know about other interventions to improve digital technology uptake in the classroom that could be alternatives to School Teams? (Is there a systematic review? Have authors reviewed the primary literature?)

Yes, we have done that. We have only found one systematic scoping review by Albers and Pattuwage, which we cite as [16] in our reference list. Four other systematic reviews take up barriers and facilitators to the uptake of digital technology, which are crucial parts of the implementation process. However, those reviews do not use implementation frameworks. So, we did not present them in the manuscript.

Quasi-experiment vs. randomisation

- In this study, schools would be self-selecting to receive the intervention, rather than randomized. This can have an effect on the results in the favour of the intervention. For instance, schools that are already motivated will be more likely to participate, and they would then be compared to schools that were not so motivated to use digital technology. Authors should acknowledge the weaknesses of a non-randomised design and describe what measures they will take to compensate. (e.g. consider before-after or time-series analysis of intervention and control group). Additionally, final results may be less likely to be applicable to schools that are not already motivated, and therefore do not sign up to participate – this should be discussed.

This is a good point. We had already considered a before-after study of the experimental schools, but we did not explain that in the manuscript. We have now added a sentence about that on page 9 and discussed it on page 30.

LAD co-design/development/evaluation

- To an outsider, it seems to me that DLM is a very broad term covering any learning material that is digital, while LAD is a specific product or solution that exists in a beta form (i.e. described “SAVis: a Learning Analytics Dashboard with Interactive Visualization and Machine Learning”). Is that correct? It needs clarifying.

The reviewer is correct about DLMs. But also LADs are general products. Sometimes those are built together in one and the same system. Those are explained on page 5 and 6. The example that we have of SAVis (page 8) is just to demonstrate what we could do in terms of developing a LAD. However, we do not have a LAD for the project, and we intend to develop one during the pilot part of the project, as explained on page 20-21. We believe that the responses to previous comments have clarified this.

- It is not clear to me why development of LAD would be done within a quasi-experiment design, and the LAD described in the article “SAVis…” appears to not be developed to the point of being suitable for use by people who are not digitally savvy (it looks like a tool for computer experts, not for teachers who are not using much digital technology). Before embarking on a multi-school experiment, my preference would have been to establish a preliminary “development” work package for LAD that included:

o Needs assessment regarding LAD

o Design development: Co-design with groups of key stakeholders (e.g. teachers, principles, school ICT administrators, maybe students and parents too?) involving several cycles of rapid iteration, user testing, and improvement of the LAD, to ensure that the solution is useable, useful, accessible, desirable, and suited/fit for use in schools’ teaching and learning environments.

o Piloting of LAD use by teachers and principles, with data collection (e.g. observation, interviews, logs)

o Assess the effect of LAD on teachers/school principles use, satisfaction, and on student outcomes if possible, by conducting a randomised trial , to understand if this innovation has the intended effect, before implementing it broadly. (You should be able to show that this LAD product has the desired effect on a set of outcomes, before setting about to implementing it on a broad scale).

The reviewer might have missed the development of the LAD on page 20-21. We co-design the LAD in the same fashion as explained by the reviewer. We have modified the text page 8 and page 20-21 to hopefully clarify the development sequence.

School team intervention – development and evaluation before implementation?

- It might be prudent to carry out a preliminary study as I have suggested above (co-development of the School Team intervention by a group of pilot schools, followed by assessment of effect using randomized trial methodology), before seeking to implement this intervention on a broad scale in a non-randomised experiment. Perhaps authors have intended this, but I have not understood it correctly.

Yes, the idea of the pilot schools to test the implementation steps is presented on page 8. We have also added a new sentence to stress this.

Reviewer #2: This paper describes a protocol that will be implemented in two areas of Sweden to document their school's digital transformation. The planned process has been thoroughly described and the research questions are clear and should provide comprehensive findings. The potential number of participants is quite large and, as such, will require detailed procedures such as outlined here.

While the protocol is well-considered, there are a number of general comments that need to be made. Overall the literature cited is dated. For example, the research that refers to 'the existing DLMs' was published in 2012. This is quite a long timeline for something as dynamic as technology implementation.

We understand the issue raised by the reviewer, however, the published article dated 2012, Siemens & Baker, is a seminal paper defining LA for the first time, and still the reference used broadly today. About the other references about DLM, we used both older and new ones to show that the evidence has not really advanced in this field, like for example references 2 and 3 on page 4.

The protocol makes little reference to the technology infrastructure roll out. I know little about the Swedish school system and it may be already very well resourced in this area and this is already in place? 'Stage 2: Installation' makes the only reference to 'Acquire resources' but there seems little mention elsewhere of infrastructure and specialist technical staff. These would be things that are likely to have a huge impact on the progress of the digital transformation of a school. There is a statement made that 'the intervention program does not require major organizational changes' so perhaps infrastructure and appropriate staffing is already in place.

Good catch! We have added a couple of sentences on this topic on page 10, under the section Schools.

I wonder if any provision has been made for cross-school comparisons when each local municipality is responsible for the purchase of resources? Will all schools be equally well-resourced in this case and how might any variations affect the research results?

Every school in Sweden has a DLM in school. Its high or low usage is mostly due to teacher competences and willingness not accessibility. This is where this project comes in. Since each municipality decides what DLM to buy, we have nothing to say about the purchase. As you rightly suggested, we will compare teachers attitudes to use of specific DLMs to help the DLM companies to further develop their product and help them develop more qualitative DLM. This will therefore be a result of the research, hopefully. 

I also note that the EdTech 'companies are full research partners and have a say on the running of the researach work'. This will have to be managed carefully to ensure there is no conflicts of interest.

The reviewer is right. We have been careful and have taken great measure in drawing a clear and standard contract with the CEOs of the EdTech Companies and the head of the department to make sure that it is clear what is expected of each other. This is mentioned at the beginning of the Edtech companies section on page 11.

This will be a significant project and the results will potentially have wide-ranging impact. I look forward to reading about it further in the future.

We do too! Thanks.

---

## [Editor Report · Decision Letter 1]

5 Dec 2023

Digital Transformation in Schools of two South Regions of Sweden through Implementation-Informed Approach: A Mixed-Methods Study Protocol

PONE-D-23-04795R1

Dear Dr. Masiello,

We’re pleased to inform you that your manuscript has been judged scientifically suitable for publication and will be formally accepted for publication once it meets all outstanding technical requirements and corrections of some minor typographical errors (see attached "reviewed manuscript with editor comments").

Kind regards,

Subhashni Taylor, PhD

Academic Editor

PLOS ONE
---

## [Editor Report · Acceptance letter]

7 Dec 2023

PONE-D-23-04795R1 

Digital transformation in schools of two southern regions of Sweden through implementation-informed approach: A mixed-methods study protocol 

Dear Dr. Masiello:

I'm pleased to inform you that your manuscript has been deemed suitable for publication in PLOS ONE. Congratulations! Your manuscript is now with our production department. 

Kind regards, 

on behalf of

Dr. Subhashni Taylor 

Academic Editor

PLOS ONE